# Type IV P-Type ATPases: Recent Updates in Cancer Development, Progression, and Treatment

**DOI:** 10.3390/cancers15174327

**Published:** 2023-08-30

**Authors:** Eugenia M. Yazlovitskaya, Todd R. Graham

**Affiliations:** Department of Biological Sciences, Vanderbilt University, Nashville, TN 37232, USA

**Keywords:** flippase, P4-ATPase, phosphatidylserine (PS), cancer development and progression

## Abstract

**Simple Summary:**

Lipids are the main structural component of cellular membranes; they play an essential role in cellular functions. For the growth and progression of cancer, tumor cells require lipid-related metabolic and structural adaptations, such as altered lipid membrane composition, which is in part regulated by lipid flippases. Type IV P-type ATPases (P4-ATPases), the main class of flippases, are responsible for membrane lipid asymmetry, and thus they are closely involved in cancer-related lipid changes. Lately, flippases have gained more attention regarding the pathobiology of human disease. This review focuses on recent publications discussing the role of P4-ATPases in cancer development, progression, and prospective treatment.

**Abstract:**

Adaptations of cancer cells for survival are remarkable. One of the most significant properties of cancer cells to prevent the immune system response and resist chemotherapy is the altered lipid metabolism and resulting irregular cell membrane composition. The phospholipid distribution in the plasma membrane of normal animal cells is distinctly asymmetric. Lipid flippases are a family of enzymes regulating membrane asymmetry, and the main class of flippases are type IV P-type ATPases (P4-ATPases). Alteration in the function of flippases results in changes to membrane organization. For some lipids, such as phosphatidylserine, the changes are so drastic that they are considered cancer biomarkers. This review will analyze and discuss recent publications highlighting the role that P4-ATPases play in the development and progression of various cancer types, as well as prospects of targeting P4-ATPases for anti-cancer treatment.

## 1. Introduction

A typical cellular membrane is a bilayered lipid structure with embedded proteins. The diversity of membrane lipids and asymmetric lipid composition of the membrane leaflets are remarkable [1]. Among major phospholipids, phosphatidylserine (PS) and phosphatidylethanolamine (PE) are mainly localized in the inner leaflet of the plasma membrane, whereas phosphatidylcholine (PC) and sphingomyelin (SM) are concentrated in the outer leaflet [2]. This membrane asymmetry is vital for cellular functions and survival. For example, the increased level of external cellular PS exposure during apoptosis provides an “eat me” signal to surrounding macrophages while suppressing an autoimmune response [3]. Cancer cells may acquire this feature to mimic the anti-inflammatory action of apoptotic cells. This adaptation results in the failure of the immune system to detect cancer cells; therefore, cancer cells are protected from the immune response and continue proliferating.

Membrane lipid asymmetry is maintained by flippase, floppase, and scramblase transmembrane enzymes. Flippases, which translocate lipids from outer to inner membrane leaflet, and floppases, which are responsible for the reverse process, utilize ATP to provide the energy needed for lipid transport. Scramblases transport lipids in both directions across the membrane and are energy-independent. Most flippases belong to the type IV P-type ATPases. P-type ATPases, a superfamily of evolutionarily related ion and lipid pumps, are named based on their ability to catalyze auto-phosphorylation (hence the designation P-type) of a key conserved aspartate residue within the pump and their energy source, ATP [4]. Based on sequence similarity, the P-type ATPases are divided into five types with different transport specificities [5]. Types I-III (P1, P2, P3) P-type ATPases catalyze cation uptake and/or efflux, while type IV P-type ATPases (P4-ATPases) flip phospholipids and type V (P5) P-type ATPases transport polyamines or mediate protein dislocation from membranes [6]. P4-ATPases are expressed in all eukaryotic cells and are the primary class of flippases responsible for establishing and maintaining membrane asymmetry. Their activities are the foundation for establishing and maintaining membrane inner and outer leaflets phospholipid asymmetry, which is essential for membrane protein regulation, polarity maintenance, signaling events, and vesicle trafficking (reviewed in [7,8,9]). While there are a number of well-structured analyses of studies demonstrating the importance of P4-ATPase functions in maintaining blood homeostasis, liver metabolism, neural development, and the immune response [10,11], no comprehensive reviews are available pertinent to the role of P4-ATPases in cancer biology. This review summarizes recent updates on type IV P-type ATPases play in cancer development, progression, and treatment.

## 2. Flipping Phospholipids in Cancer Cells

The classification of P4-ATPases is based on phylogenetic similarities of the protein sequences [11]. Classical substrates of the P4-ATPases include aminophospholipids PS and PE. Of the 15 mammalian P4-ATPases, class 1a (ATP8A1, 2) and class 6 (ATP11A, B, C) translocate both PS and PE; class 1b (ATP8B1, 3) translocate PS; and class 1b (ATP8B1, 2) and class 5 (ATP10A, B) translocate PC and/or glucosylceramide (GlcCer) (Table 1).

### 2.1. Phosphatidylserine

The asymmetry of PS across the plasma membrane plays a crucial signaling role in numerous physiological processes. In normal conditions, PS is primarily localized on the inner leaflet of the cell membranes [2]. Many membrane-associated proteins have PS-binding domains on the cytoplasmic surfaces. In the case of peripheral membrane proteins, their interaction with PS regulates their movements around a cell and their functions in different locations. For integral membrane proteins, the membrane asymmetry, including PS membrane leaflet localization, is critical for their conformation, which regulates their activity and functions [2]. Exposure of PS on the external surface of the cell membrane is a prominent feature of cells undergoing apoptosis. It marks cells to be targeted and engulfed by surrounding macrophages, since apart from other cell types, macrophages have a PS-binding site of the PS-sensing receptors in ectoplasmic domains, allowing macrophages to detect apoptotic cells. Interestingly, viable, non-apoptotic cancer cells display increased surface PS expression compared to normal cells [17]. However, macrophages do not phagocytose tumor cells because of high levels of CD47, a cluster of differentiation 47, in tumor cells, which inhibits phagocytosis vis interaction with SIRPα, signal regulatory protein α [18,19]. A combination of high PS levels on the outer leaflet of the cell membrane and high cellular CD47 levels allows cancer cells to mimic the immunosuppressive anti-inflammatory features of apoptotic cells while not being phagocytized by macrophages, which results in a high rate of cancer cell proliferation. The PS-dependent prevention of immune reaction is caused by the ligation of PS to receptors present on dendritic cells, macrophages, and T cells. The ligation of PS to receptors on macrophages promotes macrophage polarization from a proinflammatory M1-like phenotype towards a protumor M2-like phenotype, allowing the secretion of the anti-inflammatory immunosuppressive cytokines interleukin-10 and TGF-β [20]. While the exposure of PS on the extracellular surface of apoptotic cells is mainly caused by the activation of scramblase, there are data supporting a significant role of flippase deficiency for this process in cancer cells (Figure 1).

A study of abnormally high PS exposure in cancer cells of various origins revealed an inverse correlation between flippase activity and constitutive PS externalization [21]. Elevated external PS exposure in cancer cells was accompanied by high total cellular PS. In addition, steady-state calcium levels, possibly by inhibiting flippase activity, also affect PS exposure. Therefore, a high level of external PS in cancer cells seems to be a result of low flippase activity [21]. The study using tumorigenic N18 and non-tumorigenic HN2 hippocampal neuron-derived cells showed both cell lines express P-type ATPase Atp8a1, but only the neuroblastoma N18 cells use this protein as a plasma membrane flippase. Targeted knockdown of this enzyme caused PS externalization and phagocytic removal of these cells [22]. Another study demonstrated the development of PS^out^ tumor models with tumor cells lacking PS, the flippase component CDC50A, constantly exposing PS but alive [23]. Altogether, these studies suggest that insights into PS exposure mechanisms in cancer cells might facilitate tumor cell-specific induction in surface PS, and thus enhance the effectiveness of PS-targeting drugs. Therefore, flippases, responsible for PS transfer from the outer to the inner membrane surface, represent attractive anti-cancer therapeutic targets.

### 2.2. Phosphatidylethanolamine

While PS is one of the major studied membrane lipids in cancer biology, closely related aminophospholipid PE is underappreciated [24]. PE comprises 20–50% of total phospholipids, and is the second-most abundant phospholipid in mammalian cells after PC, while PS is a quantitatively minor membrane phospholipid representing 2–10% of total phospholipids [25]. In addition to PS, PE has increased surface representation on the outer membrane of tumor cells [26,27]. In addition, PE has a high degree of surface expression on endothelial cells in tumor vasculature [28], making it an attractive molecular target for future cancer interventions.

PE is highly enriched in mitochondrial inner membranes [29] suggesting the importance of PE for the Warburg effect. The Warburg effect is a classical metabolic phenotype of cancer cells that is characterized by the switch of energy production from oxidative phosphorylation in the mitochondria as observed in normal cells to a less efficient process of aerobic glycolysis even in the presence of abundant oxygen. Indeed, the Warburg effect involves a much more complex combination of multiple factors and can account for oncogenesis, tumor progression, and chemotherapy resistance of cancer cells [30]. A study using hepatocellular carcinoma cells HepG2 showed that ATP8B1 knockdown in HepG2 cells leads to a strong increase in the mitochondrial oxidative phosphorylation without a change in glycolysis, which coincided with increased mitochondrial fragmentation and phosphatidylethanolamine levels [31]. This data could be considered as a “reverse” Warburg effect, suggesting PE flippases are a potential anti-cancer molecular target.

### 2.3. Phosphatidylcholine

PC is the most abundant compound of the lipid bilayer. In normal cells, in comparison to PS and PE, it is mostly present in the outer leaflet of the plasma membrane [32]. A number of cancers exhibit the impairment of PC biosynthesis and PC metabolism alterations, resulting in a decrease in its membrane level [26]. However, the studies about specific PC distribution between inner and outer leaflets in membranes of cancer cells are very limited. Interestingly, several publications functionally connect multidrug resistance transporter (MDR1; Pgp, P-glycoprotein; ABCB1) with phosphatidylcholine flippase (MDR2) [33,34]. There is also a suggestion that MDR1 functions as a broad-specificity flippase for GlcCer, simple glycosphingolipids, and membrane phospholipids, including PC [35]. Although MDR proteins do not belong to the P4-ATPases family and are beyond the topic of this review, an association of PC membrane asymmetry with the transmembrane drug delivery system is intriguing.

## 3. P4-ATPases in Cancer Development and Progression

Recent developments in cancer research demonstrate the involvement of P4-ATPases in the development and progression, including metastases, of various types of cancers (Table 2). Most information is related to class 1 and 6 flippases.

### 3.1. Gastrointestinal Cancers

For colorectal cancer (CRC), the functional role of various P4-ATPases is highly controversial depending on the specific study. *ATP8B1* was initially identified as a driver gene for sporadic CRC by Genomic Identification of Significant Targets In Cancer (GISTIC) [44]. Using three-level exon array data (gene, exon, and network) and additional separate differential expression analyses, Significance Analysis of Microarrays (SAM), and Linear Models for MicroArray data (LIMMA), *ATP8B1* was found to be a novel gene associated with CRC that shows changes at cytogenetic, gene and exon levels [45]. Additionally, analyses of transcriptomics, genomics, and clinical data of CRC samples from The Cancer Genome Atlas (TCGA) revealed that ATP8B1 was the only potential biomarker of phospholipid transporters in CRC. Its prognostic value was also validated with the data from the Gene Expression Omnibus (GEO) [46]. However, in another study, ATP8B1 was suggested as a tumor suppressor for CRC since the forced reduction in ATP8B1 expression either by CRISPR/Cas9 or shRNA was associated with increased growth and proliferation of the CRC cell line HT29, while an overexpression of ATP8B1 resulted in reduced growth and proliferation of the SW480 CRC cell line [47]. Gene expression for another P4-ATPase, *ATP11A*, was assessed in 7 colorectal cancer cell lines and 95 paired cases of colorectal cancer and non-cancerous regions. A high level of *ATP11A* was proposed as a novel predictive marker for metachronous metastasis of colorectal cancer [62]. In addition, epigenome-wide methylation analysis of CpG sites of genes in visceral adipose tissue of CRC patients vs. healthy individuals detected a correlation between high methylation levels of *ATP11A* with CRC [63].

A deficiency of P4-ATPase ATP8B1 is mainly associated with progressive familial intrahepatic cholestasis type 1. However, a next-generation transcriptome sequencing study of three human hepatocellular carcinoma (HCC) tumor/tumor-adjacent pairs, followed by validation of differential gene expression findings in a large data set consisting of 434 liver normal/tumor sample pairs, revealed non-synonymous mutations in ATP8B1 as liver cancer driver mutations [48]. On the other hand, for patients with HCC, high ATP9A levels predicted a poor outcome [73]. In hepatocellular carcinoma cells, ATP9A was critical for regulating macropinocytosis via interaction with ATP6V1A resulting in plasma membrane cholesterol accumulation. Macropinocytosis, an important process that fuels cancer cell proliferation by scavenging extracellular proteins or necrotic cell debris, is critical for sustaining HCC proliferation and growth under nutrient limitation. The interaction of ATP9A with ATP6V1A facilitated ATP6V1A membrane trafficking, thereby activating plasma membrane cholesterol-dependent RAC1 signaling and initiating macropinocytosis to increase the energy supply and support HCC cell survival and proliferation [73]. Another study detected that miR-103a was one of the most highly expressed microRNAs in HCC tissues and that a high expression of miR-103a was associated with poor patient prognosis. Experiments using HCC cells in culture and subcutaneous xenograft models showed that miR-103a promoted glucose metabolism and directly inhibited cell death by targeting ATP11A and EIF5 transcripts, therefore, suggesting a regulatory mechanism(s) for HCC progression [64].

*ATP8B2* was identified as one of the nine macrophage phenotypic switch-related genes (MRGs) with satisfactory prognostic ability in multiple analyses of databases from the Cancer Genome Atlas (TCGA)-pancreatic adenocarcinoma (PAAD), Genotype-Tissue Expression (GTEx)-Pancreas, and Gene Expression Omnibus (GEO) [51]. The critical role of *ATP8B2* as one of four N6-methyladenosine (m^6^A)-associated metabolic genes and prognostic biomarkers of the immune response in cancer progression was confirmed in a study of pancreatic ductal adenocarcinoma. Overall, amplification was the most frequent type of mutation in *ATP8B2*, as well as the highest rate of genetic alterations (4%) among these four genes [52]. In a comparison of pancreatic cancer and paracancerous tissues, ATP11A mRNA and protein levels were significantly higher in cancer [65]. Moreover, ATP11A promoted the invasion and migration of cultured pancreatic cancer cells via TGFB-dependent epithelial-mesenchymal transition (EMT) [65].

### 3.2. Prostate Cancer

Most recently, ATP8B1 was identified as a prognostic prostate cancer biomarker. The investigated association of 222 haplotype-tagging SNPs in eight phospholipid-transporting ATPase genes with cancer-specific survival and overall survival of 630 patients treated with androgen-deprivation therapy (ADT) for prostate cancer revealed that ATP8B1 was under-expressed in tumor tissues, and that a higher ATP8B1 expression was associated with a better patient prognosis [49]. Specifically, *ATP8B1* rs7239484 was proven useful in predicting the efficacy of ADT. Another study discovered *ATP11A* as a biomarker that improved discrimination of patients with metastatic-lethal prostate cancer. In this study, epigenome-wide DNA methylation profiling of surgically resected primary tumor tissues was compared between a population-based and a replication cohort of prostate cancer patients with five years follow-up [66]. ATP11A DNA methylation levels were significantly correlated with transcript levels and predicted metastatic-lethal events in the validation cohort. High methylation levels of CpGs in ATP11A also correlated with loss of the tumor suppressor gene *PTEN*, which has an unfavorable prognosis in prostate cancer [67].

### 3.3. Endometrial, Cervical, Ovarian and Breast Cancers

In a study of exome-wide rare variant association with endometrial cancer using germline whole exome data from the MyCode community health initiative, *ATP8A1* was identified as one of several candidate genes involved in endometrial cancer predisposition, which could help in personalized prognosis [36].

A recent study using integrated bioinformatics analysis for expression and methylation of exocytosis genes from The Cancer Genome Atlas Cervical Squamous Cell Carcinoma and Endocervical Adenocarcinoma (TCGA-CESC) dataset identified *ATP8B4* as one of nine genes linked with metastasis in cervical carcinoma [53]. Another study on integrative analysis of the DNA methylome and transcriptome in uterine leiomyoma found hypomethylation/upregulation of *ATP8B4* compared with adjacent myometrium [54]. These studies speculated that reversion of this methylation could offer a therapeutic option for these types of cancer. In a genome-wide DNA methylation profiling study of cervical cancer stages, in which CpG sites whose state of methylation correlated with lesion grade were assessed, *ATP10A* was short-listed as a gene associated with the progression from normal through precancerous lesions and cervical cancer [57].

For ovarian cancer, *ATP11B* high expression was found to correlate with higher tumor grade in human ovarian carcinoma samples and with cisplatin resistance in human ovarian cancer cell lines [69]. *ATP11B* gene silencing restored the sensitivity of ovarian cancer cell lines to cisplatin, as well as in mice bearing ovarian tumors derived from cisplatin-sensitive and resistant cells. Mechanistic studies in cell lines suggested that ATP11B contributes to the secretory vesicular transport of cisplatin from the Golgi to the plasma membrane [69]. In contrast, another study reported that when relative gene expression in a set of primary epithelial ovarian cancer and control ovarian tissues were compared with clinical data and survival of patients, levels of *ATP11B* were decreased in carcinomas [70].

Data from three genome-wide association studies (GWAS) were reanalyzed using a novel computational biostatistics approach (muGWAS) for validation of single nucleotide polymorphisms associated with incurable metastatic breast cancer [37]. Findings from this study included ATP8A1 and ATP8B1 among other proteins that translocate and metabolize phospholipids, which control endo-/exocytosis. These novel findings suggest scavenging phospholipids as a novel intervention to control the local spread of cancer, packaging of exosomes, and endocytosis of specific receptors [37]. Another P4-ATPase implicated in breast cancer is ATP8A2. Differential expression analysis with the subsequent univariate Cox regression and LASSO algorithm were used to uncover key prognostic genes for 611 luminal A breast cancer patients [40]. This analysis resulted in a 5-gene-risk score model for predicting luminal A-invasive lobular breast cancer survival. Specifically, 105 prognostic genes and 9 predictors were identified, allowing the identification of 5-key prognostic genes including *ATP8A2* [40]. In another study, analysis of the cells from breast cancer patients demonstrated that no or low expression of ATP11B in conjunction with high expression of PTDSS2, which is negatively regulated by BRCA1, markedly accelerates tumor metastasis and associates with poor prognosis [71].

### 3.4. Blood Cancers

An under-expression of *ATP9A* was reported in relapsed follicular lymphoma patients upon the analysis of 38 differentially expressed ion channels and transporter genes identifying *ATP9A* as one of the novel lymphoma biomarkers related to excitability and metabolic pathways, with particular relevance for drug-resistant, relapsed follicular lymphoma [56].

While childhood acute lymphoblastic leukemia (ALL) typically has a favorable prognosis, a substantial subset of patients with childhood ALL frequently relapses. Recurrent aberrations of 14 genes in patients who subsequently relapsed were detected, including deletions/uniparental isodisomies of ATP10A in B-cell precursor ALL [58]. Additionally, enriched mutations, although infrequent, in ATP10A were linked to relapsed chronic lymphocytic leukemia (CLL) [59]. ATP11A levels of long non-coding RNA were significantly upregulated in ALL [74]. Furthermore, low levels of *ATP11A* expression/methylation were identified as independent prognostic factors for another blood malignancy, acute myeloid leukemia (AML) using the methylation array data and mRNA array data from the Gene Expression Omnibus (GEO) database [68].

### 3.5. Lung Cancers

The role of ATP8A1 in the development and progression of non-small-cell lung cancer (NSCLC) was discovered in a study of the tumor suppressor function of microRNA MiR-140-3p. The growth of NSCLC cells in nude mouse models was suppressed by an overexpression of miR-140-3p, which was attenuated by an overexpression of ATP8A1 [39]. In a study of 25 cases of tumor tissues and the adjacent normal tissues from surgeries of NSCLC patients, ATP8A1 was found overexpressed in NSCLC tissues by immunohistochemical staining [38]. Mechanistically, ATP8A1 promoted the expression of MMP-9 and Vimentin, as well as suppressed the expression of E-cadherin, thus resulting in the elevated invasion/migration ability of NSCLC cells [38].

A study of differential gene expression analysis revealed that *ATP8A2* and four other genes, each harboring one of the five frequently hypermethylated CpG sites within its promoters, were frequently downregulated in lung adenocarcinoma (LUAD) [41]. Poor expression of *ATP8A2* in LUAD was also discovered in the analysis of the LUAD-related mRNA expression profile using univariate and multivariate Cox regression [43]. Additionally, *ATP8A2* was found as 1 of 18 prognostic biomarkers in an analysis of the integrated relationship between nuclear mitochondrial genes (NMGs) and the progression of lung adenocarcinoma [42]. Machine learning algorithms were used to investigate the gene expression profiles of lung adenocarcinoma (AC) and lung squamous cell cancer (SCC) samples retrieved from Gene Expression Omnibus [72]. In this study, high levels of *ATP11B* were specifically detected in AC, but not in SCC.

ATP8B1 was proposed as a novel predictive biomarker in lung squamous cell carcinoma (LUSC). Investigation of the whole genomic expression profiles of LUSC samples from The Cancer Genome Atlas (TCGA) database and Tianjin Medical University Cancer Institute and Hospital (TJMUCH) cohort revealed that the low expression of ATP8B1 was associated with poor prognosis of LUSC patients [50]. In LUSC cell lines, ATP8B1 knockdown using lentiviral infection with shRNA promoted proliferation, inhibited apoptosis, and aggravated invasion and migration [50].

ATP10D was specifically identified as a positive biomarker in individuals presenting extreme phenotypes of low risk of developing tobacco-induced lung cancer [61]. SNPs associated with the risk of developing tobacco-induced NSCLC were analyzed in independent discovery and validation sets of patients followed by assessment of the prognostic value of mRNA expression and protein expression of target genes. A significant correlation of low mRNA expression of ATP10D with shorter survival in patients with stage I–II NSCLC confirmed the prognostic value of ATP10D [61].

### 3.6. Melanoma

The whole-exome sequencing of 34 melanoma-prone families (119 cases) coupled with coexpression network analyses focused on modules associated with pigmentation processes introduced 36 genes (including *ATP10A)* as potential melanoma risk genes in the families [60].

Hepatocyte growth factor-overexpressing mice that harbor a deletion of the Ink4a/p16 locus (HP mice) form melanomas with low metastatic potential in response to UV irradiation. However, these tumors become highly metastatic following hemizygous deletion of the Nme1 and Nme2 metastasis suppressor genes (HPN mice) [55]. Interestingly, missense mutations in eight signature genes including Atp8b4 were associated with a striking increase in lung metastatic activity in the melanoma model of HPN mice. Analysis of transcriptome data from The Cancer Genome Atlas revealed that expression profiles of these genes may serve as prognostic markers and/or therapeutic targets for clinical management of metastatic melanoma [55].

## 4. P4-ATPases as Targets for Anti-Cancer Treatment

### 4.1. Interplay with the Small Molecule Inhibitors

ATP8B3 was proposed as one of eight genes that regulate oxaliplatin response in the treatment of CRC. The presence of ATP8B3 SNPs correlated with the increased sensitivity of CRC to oxaliplatin therapy. Its predictive value was confirmed in a two-step procedure to comprehensively investigate 1444 single nucleotide polymorphisms (SNPs) from metabolic and transporter enzymes, using 623 stage II–IV CRC patients (of whom 201 patients received oxaliplatin) from a German prospective patient cohort treated with adjuvant or palliative chemotherapy [75]. It was further validated in a follow-up study of 53 SNPs using data from 1502 patients with stage II-IV colorectal cancer who received primary adjuvant chemotherapy and confirmed as predictive functional SNP with significance at *p* < 0.05 [76].

Greatly increased levels of ATP11A were detected in a study using breakpoint cluster region/Abelson murine leukemia (Bcr/Abl) P190 lymphoblasts with resistance to farnesyltransferase inhibitors (FTI) [77]. The same study demonstrated that the overexpression of *ATP11A* provided protection against the effects of FTI SCH66336, whereas knockdown of endogenous ATP11A using small interfering RNA (siRNA) made cells more sensitive to this drug. The lymphoblasts that were resistant to SCH66336 were also more resistant to structurally similar FTIs, FTI-276 and GGTI-298, and to imatinib mesylate, the Bcr/Abl tyrosine kinase inhibitor. Since elevated levels of ATP11A can protect malignant lymphoblastic leukemia cells against several novel small molecule signal transduction inhibitors, therefore, ATP11A could serve as a target for chemotherapy effects [77].

MiR-140-3p has a tumor suppressor function via inhibition of cancer cell growth, migration, and invasion. ATP8A1 was demonstrated as a novel direct target of miR-140-3p [39]. NSCLC cells in nude mouse models were suppressed by the overexpression of miR-140-3p. The increased level of intracellular ATP8A1 protein attenuated the inhibitor role of miR-140-3p suggesting ATP8A1 as a potential target for anti-cancer combination treatment.

### 4.2. Immunotherapy

In cancer treatment, oncolytic immunotherapy is one of the most promising approaches to boost antitumor immune response. The local injection of oncolytic peptides can induce immunogenic cell death leading to the activation of the immune system and the enhancement of the immune response [78]. One such peptide, LTX-315 (oncolytic peptide deriving from bovine lactoferrin), downregulates the expression of PD-L1 (programmed cell death ligand 1), a representative immune-inhibitory checkpoint molecule, and enhances CD8+ T cell infiltration in pancreatic cancer [79]. ATP11B was identified as a potential target of LTX-315 and a critical regulator in maintaining PD-L1 expression in pancreatic cancer cells and a mouse model [79]. Mechanistically, ATP11B interacted with PD-L1 in a CKLF-like MARVEL transmembrane domain containing 6 (CMTM6)-dependent manner. The depletion of ATP11B promoted CMTM6-mediated lysosomal degradation of PD-L1, thus reactivating the immune microenvironment and inducing an antitumor immune response. The significant correlation between ATP11B and PD-L1 was confirmed in clinical samples of pancreatic cancer. Since PD-L1 largely determines the efficacy and effectiveness of cancer immunotherapies, the development of ATP11B-targeting drugs is essential for anti-pancreatic cancer immunity.

The macrophage phenotype switch plays a vital role in the progression of malignancies. As mentioned previously, *ATP8B2* was detected as one of nine signature macrophage phenotypic switch-related genes that could be used as an independent prognostic index for pancreatic adenocarcinoma (PDAC) patients [51]. Macrophages are one of the most abundant immune cells in PDAC tumor microenvironment. According to their polarization states, macrophages are roughly categorized into two types: classically activated type 1 (M1 macrophages), and alternatively activated type 2 (M2 macrophages). M2 macrophages exert pro-tumor functions, whereas M1 macrophages exert anti-tumor functions [80]. The macrophage phenotypic switch-related genes indicate a change in macrophage type and provide prognostic information for PDAC treatment. Later, *ATP8B2* was confirmed as part of a four-gene signature to predict the immune response to treatment and overall survival of pancreatic adenocarcinoma patients [52] suggesting ATP8B2 as a target for anti-cancer immunotherapy.

## 5. Conclusions and Future Directions

This review discussed recent literature identifying P4-ATPase alterations associated with cancer predisposition, association, development, and progression. These alterations occur at the genomic, translational, and transcriptional levels, reflecting the variety of possible flippase-dependent regulatory mechanisms involved in cancer pathobiology. Much progress has been made in the P4-ATPase field in the last few years. However, there is still a need for additional research directed towards molecular mechanisms and the development of flippase-targeted anti-cancer therapeutic interventions.

## Figures and Tables

**Figure 1 cancers-15-04327-f001:**
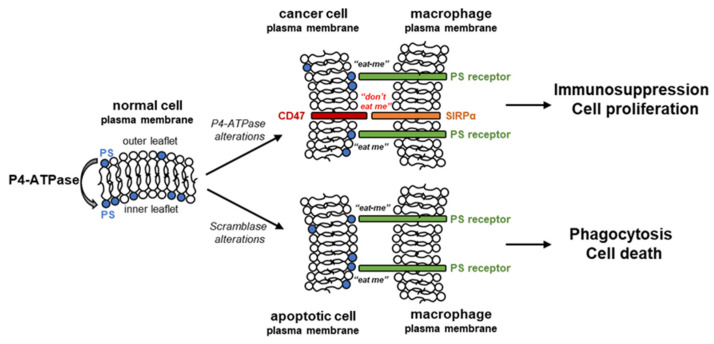
Altered phosphatidylserine plasma membrane leaflet localization in cancer and apoptotic cells determines cell fate. PS, phosphatidylserine; SIRPα, signal regulatory protein α.

**Table 1 cancers-15-04327-t001:** Substrate specificities of mammalian P4-ATPases.

Class	Name	Lipid Substrate	Sub-Cellular Localization	References
1a	ATP8A1	PS, PE	PM, TNG, endosome	[12]
1a	ATP8A2	PS, PE	PM, TNG, endosome	[12]
1b	ATP8B1	PS, PC, cardiolipin?	PM	[12,13]
1b	ATP8B2	PC	PM	[12,13]
1b	ATP8B3	PS?	ER, TNG	[12,14,15]
1b	ATP8B4	unknown	PM	[12]
2	ATP9A	unknown	TNG, endosome	[16]
2	ATP9B	unknown	TNG	[16]
5	ATP10A	PC, GlcCer	PM	[16]
5	ATP10B	PC, GlcCer	endosome, lysosome	[16]
5	ATP10D	GlcCer	PM	[16]
6	ATP11A	PS, PE	PM	[13,16]
6	ATP11B	PS, PE	endosome	[13,16]
6	ATP11C	PS, PE	PM	[13,16]

**Table 2 cancers-15-04327-t002:** P4 ATPases in cancer development and progression.

ATPase	Cancer Type	Reported Feature	Reported Effect	Citations
ATP8A1	endometrial cancer	gene expression up	cancer predisposition	[36]
	metastatic breast cancer	SNPs	cancer association	[37]
	non-small-cell lung cancer	protein expression up	cancer association	[38]
	non-small-cell lung cancer (mouse model)	protein expression up	cancer cell growth	[39]
ATP8A2	luminal A-invasive lobular breast cancer	gene expression up	poor prognosis	[40]
	lung adenocarcinoma	gene expression down	cancer association	[41,42]
	lung adenocarcinoma	mRNA expression down	cancer association	[43]
ATP8B1	colorectal carcinoma	gene expression up	cancer association	[44,45,46]
	colorectal carcinoma (cell lines)	protein expression down	increased cell growth	[47]
		protein expression up	decreased cell growth	
	hepatocellular carcinoma	gene mutation	cancer association	[48]
	prostate cancer	protein expression up	better prognosis	[49]
	metastatic breast cancer	SNPs	cancer association	[37]
	lung squamous cell carcinoma	protein expression down	poor prognosis	[50]
ATP8B2	pancreatic adenocarcinoma	gene expression up	poor prognosis	[51]
	pancreatic ductal adenocarcinoma	gene expression up	poor prognosis	[52]
ATP8B4	metastasis in cervical carcinoma	gene methylation down	cancer association	[53]
	uterine leiomyoma	gene methylation down	cancer association	[54]
	metastasis in melanoma (mouse model)	gene mutations	cancer progression	[55]
ATP9A	hepatocellular carcinoma	protein expression up	poor outcome	[43]
	relapsed follicular lymphoma	gene expression down	poor prognosis	[56]
ATP10A	cervical cancer	gene methylation up	cancer progression	[57]
	relapsed acute lymphoblastic leukemia	gene mutations	poor prognosis	[58]
	relapsed chronic lymphocytic leukemia	gene mutations	poor prognosis	[59]
	melanoma	gene expression up	cancer predisposition	[60]
ATP10D	tobacco-induced lung cancer	mRNA expression down	poor prognosis	[61]
ATP11A	metastasis of colorectal carcinoma	gene expression up	cancer association	[62]
	colorectal carcinoma	gene methylation up	cancer association	[63]
	hepatocellular carcinoma (mouse model)	mRNA expression down	cancer progression	[64]
	pancreatic cancer	protein expression up	cancer association	[65]
	metastatic-lethal prostate cancer	gene methylation up	poor outcome	[66,67]
	acute lymphoblastic leukemia	non-coding RNA up	cancer association	[23]
	acute myeloid leukemia	gene methylation down	poor outcome	[68]
ATP11B	ovarian carcinoma	protein expression up	cancer progression	[69]
	ovarian carcinoma	gene expression down	tumor association	[70]
	breast cancer	protein expression down	poor prognosis	[71]
	lung adenocarcinoma	gene expression up	cancer association	[72]

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
