# Peer review of "Type IV P-Type ATPases: Recent Updates in Cancer Development, Progression, and Treatment"

_cancers, 2023, doi:10.3390/cancers15174327_

Round 1

Author Response

Article: Type IV P-type ATPases: Recent Updates in Cancer Development, Progression and Treatment

Response to Reviewer 1 Comments

The authors thank reviewer 1 for insightful and helpful review. Below are point-by-point responses to the specific comments. Changes to the original manuscript are underlined in the revised version.

Point 1: L35-37. This sentence looks very interesting, but then is not connected in the rest of the

section. The authors talk again about this issue in the next section but also without conclusion. Consider to either remove from here or to elaborate more to give a general conclusion of this issue here or move these observations to the 4.2 section.

Response 1: The original statement was revised; a general conclusion was included. L36-39 in the revised version.

Cancer cells may acquire this feature to mimic the anti-inflammatory action of apoptotic cells. This adaptation results in failure of the immune system to detect cancer cells; therefore, cancer cells are protected from the immune response and continue proliferating.

Point 2: L42. “P4-ATPases” This is the first mention to the family, please mention before that is a flippasa. Is there other P1-P3 members? P4 are present only in mammals. Please rewrite for improve clarity.

Response 2: Additional information on P4-ATPases family was included. L44-52 in the revised version.

Most flippases belong to the type IV P-type ATPases. P-type ATPases, a superfamily of evolutionarily related ion and lipid pumps, are named based upon their ability to catalyze auto-phosphorylation (hence the designation P type) of a key conserved aspartate residue within the pump and their energy source, ATP [4]. Based on sequence similarity, the P-type ATPase are divided into five types with different transport specificities [5]. Types I-III (P1, P2, P3) P-type ATPases catalyze cation uptake and/or efflux, while type IV P-type ATPases (P4-ATPases) flip phospholipids, and type V (P5) P-type ATPases transport polyamines or mediate protein dislocation from membranes [6].

Point 3: L4-52. The ideas here seem contradictory, “there is limited information” but this review summarized the “recent updates”.

Response 3: The statement was clarified. L57-61 in the revised version.

While there is a number of well-structured analyses of studies demonstrating the importance of P4-ATPase functions in maintaining blood homeostasis, liver metabolism, neural development, and the immune response [10, 11], no comprehensive reviews are available pertinent to the role of P4-ATPases in cancer biology.

Point 4: L54, what are the bases of this classification? Substrate specificity?

Response 4: The required information was included. L64-65 in the revised version.

The classification of P4-ATPases is based on phylogenetic similarities of the protein sequences.

Point 5: L67-75. Although very interesting, it is difficult to get the idea. As described here, then the important thing in cancer is the presence of CD47 and not that of flippases. Please organize better the structure of the paragraph.

Response 5: The statement was clarified; an explanatory Figure was included. L83-101 in the revised version.

However, macrophages do not phagocytose tumor cells because of high levels of CD47, cluster of differentiation 47, in tumor cells, which inhibits phagocytosis vis interaction with SIRPα, signal regulatory protein α [18, 19]. A combination of high PS levels on the outer leaflet of the cell membrane and high cellular CD47 levels allows cancer cells to mimic the immunosuppressive anti-inflammatory features of apoptotic cells while not being phagocytized by macrophages, which results in high rate of cancer cell proliferation. The PS-dependent prevention of immune reaction is caused by ligation of PS to receptors present on dendritic cells, macrophages and T cells. The ligation of PS to receptors on macrophages promotes macrophage polarization from a proinflammatory M1-like phenotype towards a protumor M2-like phenotype, allowing the secretion of the anti-inflammatory immunosuppressive cytokines interleukin-10 and TGF-β [19]. While the exposure of PS on the extracellular surface of apoptotic cells is mainly caused by the activation of scramblase, there is data supporting a significant role of Flippase deficiency for this process in cancer cells (Figure).

Figure. Altered phosphatidylserine plasma membrane leaflet localization in cancer and apoptotic cells determines cell fate. PS, phosphatidylserine; SIRPα, signal regulatory protein α.

Point 6: L79-82. Please rewrite to explain that in cancer cells the activity of the ezyme is low.

Response 6: The explanatory sentence was added. L106-107 in the revised version.

Therefore, high level of external PS in cancer cells seems to be a result of low flippase activity.

Point 7: L88-90. This is a hypothesis of the authors? Please explain about the mechanisms and provide references.

Response 7: Based on the studies (references 21, 22 and 23 in the revised version) described in the mentioned paragraph (L102-117 in the revised version), the authors propose a summary for the mechanistic involvement of the fliappases in regulation of high PS exposure on the surface of cancer cells. This summary suggests flippases as the attractive anti-cancer therapeutic targets.

Point 8: Table 2:

This table is an excellent idea as intended to be abstract of the information describe in the section 3, and also for section 4. As described in all the section there are a lot of information and sometimes the results are contradictory. Since this is the aim of the review, please consider to clarify better the reported feature column: for example, in gene expression, is it increasing or going down?

Some enzymes are not included, although then are discussed in the next section, for example ATP8B3. Please correct and include in this table all the enzymes. This could facilitate to pick up the information.

Response 8: The “Reported feature” column was clarified: for protein, gene and RNA expression, “up” or “down” was added to indicate increased or decreased levels.

The intended purpose of Table 2 is to summarize and clarify studies presented specifically in section 3 (hence the title: P4 ATPases in cancer development and progression), since “there are a lot of information and sometimes the results are contradictory”, as mentioned by reviewer. Therefore, Table 2 includes information about flippases based on studies described in section 3. Studies described in section 4 (including information on ATP8B3) are not part of Table 2, since section 4 deals with targeting flippases for anti-cancer treatment. The authors feel that adding information from section 4 to Table 2 will make Table 2 less helpful in navigating complex studies presented in section 3.

Point 8: L134 GI cancer: indicates the full name.

Response 8: The full name was added. L158 in the revised version.

Gastrointestinal Cancers

Point 9: L311. Please explain if the enzyme levels are high or not, and which could be the connections with oxaliplatin.

Response 9: The explanatory sentence was added. L338-339 in the revised version.

The presence of ATP8B3 SNPs correlated with the increased sensitivity of CRC to oxaliplatin therapy.

Point 10: L320-329. Are the studies described here included in table 2? Please correct.

Response 10: Please, see Response 8.

Point 11: L346, please describe briefly the hypothesis about how ATP11b could support PDL-1 expression.

Response 11: The brief description was added. L372-376 in the revised version.

Mechanistically, ATP11B interacted with PD-L1 in a CKLF-like MARVEL transmembrane domain containing 6 (CMTM6)-dependent manner. The depletion of ATP11B promoted CMTM6-mediated lysosomal degradation of PD-L1, thus reactivating the immune microenvironment and inducing an antitumor immune response.

Point 12: L350. ATP8B levels/ expression increases or diminishes? Please indicate how this could alter macrophage function.

Response 12: The mentioned study describes the presence of mutations in ATP8B2. The clarification was added. L206-208 in the revised version.

Overall, amplification was the most frequent type of mutation in ATP8B2 as well as the highest rate of genetic alterations (4%) among these four genes [50].

Explanation on role of macrophage phenotypic switch related genes as prognostic factors for pancreatic adenocarcinoma treatment was added. L383-389 in the revised version.

Macrophages are one of the most abundant immune cells in PDAC tumor microenvironment. According to their polarization states, macrophages are roughly categorized into two types: classically activated type 1 (M1 macrophages), and alternatively activated type 2 (M2 macrophages). M2 macrophages exert pro-tumor functions, whereas M1 macrophages exert anti-tumor functions [78]. The macrophage phenotypic switch related genes indicate change of macrophages type and provide prognostic information for PDAC treatment.

Reviewer 2 Report

In this review article, the authors discussed the role of P4-ATPases in different cancers. This excellent review discusses the recent papers highlighting the role of P4-ATPases. I suggest the authors include a cartoon on phospholipid pathways and how it is altered in cancer cells. This will be a visual summary of sections 1-3 and help the readers understand the topic better.

Author Response

Article: Type IV P-type ATPases: Recent Updates in Cancer Development, Progression and Treatment

Response to Reviewer 2 Comments

The authors thank reviewer 2 for insightful and helpful review. Below is a response to the specific comment. Changes to the original manuscript are underlined in the revised version.

Point 1: I suggest the authors include a cartoon on phospholipid pathways and how it is altered in cancer cells. This will be a visual summary of sections 1-3 and help the readers understand the topic better.

Response 1: The explanatory figure was added. L98-101 in the revised version.

Figure. Altered phosphatidylserine plasma membrane leaflet localization in cancer and apoptotic cells determines cell fate. PS, phosphatidylserine; SIRPα, signal regulatory protein α.

Reviewer 3 Report

This is a timely review article dealing with the role of flippases in cancer pathologenesis. The contribution should be of value to investigators in cancer research.

Author Response

Article: Type IV P-type ATPases: Recent Updates in Cancer Development, Progression and Treatment

Response to Reviewer 3 Comments

The authors thank reviewer 3 for insightful and helpful review.